# VDR Regulates BNP Promoting Neurite Growth and Survival of Cochlear Spiral Ganglion Neurons through cGMP-PKG Signaling Pathway

**DOI:** 10.3390/cells11233746

**Published:** 2022-11-23

**Authors:** Xinyu Zhang, Ke Zhou, Keyong Tian, Qingwen Zhu, Wei Liu, Zhenzhen Liu, Xiaogang An, Chaoyong Tian, Yao Li, Fei Lu, Fei Sun, Dingjun Zha

**Affiliations:** 1Department of Otolaryngology-Head and Neck Surgery, Xijing Hospital, Air Force Military Medical University, Xi’an 710032, China; 2Department of Laboratory Medicine, Institute of Clinical Laboratory Medicine of PLA, Xijing Hospital, Air Force Military Medical University, Xi’an 710032, China

**Keywords:** sensorineural hearing loss, vitamin D receptor, brain natriuretic peptide, spiral ganglion neurons, neuronal regeneration therapy

## Abstract

Spiral ganglion neurons (SGNs) are important for hearing, and their peripheral and central processes connect sensory cells of the Corti organ to the central nervous system. The resulting network forms a point-to-point auditory conduction. As a cardiac hormone, brain natriuretic peptide (BNP) binds to natriuretic peptide receptor type A leading to diuresis, vasodilatation, inhibition of renin and aldosterone production, and cardiac and vascular myocyte growth. This study primarily aimed to explore the expression and function of BNP in the rat’s inner ear and elucidate its regulatory mechanism. We determined the expression and function of BNP and found that the vitamin D receptor (VDR) could upregulate the expression of BNP and enhance its function. In SGNs of the rat inner ear, BNP promotes neuron survival and prolongs neurite length through the cGMP-PKG signaling pathway, which could be regulated by VDR and provide a novel approach for neuronal regeneration therapy. To the best of our knowledge, this is the first study to report this potential transcriptional regulatory relationship and will act as a reference for research on neuronal regeneration therapy for SGNs injury.

## 1. Introduction

Hearing loss is a major health concern worldwide that adversely affects the lives of millions of people. Spiral ganglion neurons (SGNs) are primary neurons that carry auditory information and transmit signals from hair cells to the auditory center after initial coding processing [1]. Many factors, such as noise, infection, ototoxic drugs, and aging, directly or indirectly damage SGNs, leading to sensorineural hearing loss (SNHL). However, the regenerative capacity of the SGNs is poor. Protecting SGNs or repairing damaged neurons is crucial for rehabilitating the aural ability [2].

Brain natriuretic peptide (BNP), a member of the natriuretic peptide family that is secreted primarily by the heart, is a polypeptide consisting of 32 amino acid residues that was first detected in the brains of pigs [3]. Similar to atrial natriuretic peptide (ANP), BNP binds to natriuretic peptide receptor type A (NPRA), increasing the level of intracellular cyclic guanosine monophosphate (cGMP). This can lead to diuresis, vasodilatation, inhibition of renin and aldosterone production, and inhibition of cardiac and vascular myocyte growth [4]. Notably, BNP binds to the receptors located in the kidney, produces renoprotective effects [5], and has the ability to stimulate a physiological autocrine loop at the level of the bronchial wall, leading to bronchial dilation and protection of bronchial hyperresponsiveness [6]. The expression of BNP in neural retinal, glial, and vascular elements of the normal adult retina suggests that these peptides also play a role in maintaining the nerve and vascular integrity of the mature retina [7].

Promoting SGN regeneration and inducing neurite growth in SGNs are important areas in the field of hearing research. As the first member of the natriuretic peptide family, ANP is expressed in the rat SG and promotes SGN neurite growth in a dose-dependent manner through the NPRA/cGMP/PKG pathway [8,9,10]. On this basis, it was assumed that BNP would also have a potential effect on the inner ear and has the potential to maintain the normal function of neurons.

The vitamin D receptor (VDR) is an ancient member of the nuclear receptor superfamily and is highly conserved in birds, fish, and mammals [11,12]. VDR is predominantly distributed in the cytoplasm, where it interacts with the bioactive form of vitamin D, 1,25 (OH)_2_D_3_, heterodimerizes with the retinoid X receptor (RXR), and translocates to the nucleus. After binding with other transcription factors, VDR interacts with vitamin D-responsive elements and up- or down-regulates hundreds of genes directly controlled by vitamin D. In the mouse inner ear, mutations in the VDR gene can cause progressive deafness [13]. However, the mechanism underlying this phenomenon remains vague. In this study, we aimed to describe the roles of VDR and BNP in the inner ear and clarify the relationship between the two molecules. We hypothesized that VDR would regulate BNP, playing an important physiological function in the inner ear. To the best of our knowledge, our study is the first to illustrate that VDR influences the survival of SGNs, serving as a transcriptional regulator. The pathway through which BNP functions has also been elucidated. We expect the findings of this study to provide novel ideas for neuronal regeneration therapy for SNHL.

## 2. Methods

### 2.1. Animals and Tissue Preparation

All animals were purchased from the Experimental Animal Center of Fourth Military Medical University. All experimental protocols met the requirements and were approved by the Institutional Animal Protection and Utilization Committee of the Fourth Military Medical University. All efforts were made to minimize suffering and to reduce the number of animals used. The cochlea used in this study was obtained from postnatal day 0 (P0) to postnatal day 28 (P28) SD rats. All rats were decapitated, and their skulls opened in the middle. Under an anatomical microscope (SZX16, Olympus, Tokyo, Japan), a rat cochlea was removed from its temporal bone and bathed in cold Hank’s balanced salt solution (HBSS; H1025, Solarbio, Beijing, China) and collected for further use.

### 2.2. Plasmid and Lentivirus

The GFP-tagged VDR-overexpression plasmid, VDR-overexpression lentiviral vector (LV-VDR), and negative control lentiviral vector (LV-NC) were purchased from GenePharma (Suzhou, China). The lentiviral vector was transfected into the SGNs of cochlear explants using polybrene. The transfected explants were incubated at 37 °C for 48–72 h in a 5% CO_2_ incubator, and green fluorescence was observed. The explants were harvested, and total protein was obtained from cultured cochlear explants for further assays.

### 2.3. Cell Culture and Transfection

Next, 293T cells were cultured in Dulbecco’s modified Eagle’s medium (DMEM; C11995500BT, Gibco, New York, NY, USA), supplemented with 10% fetal bovine serum (04-001-1A, Biological Industries, Beit-Haemek, Israel) and 1% penicillin-streptomycin (P1400, Solarbio, Beijing, China). Cells were maintained at 37 °C in a 5% CO_2_ humidified atmosphere and seeded onto the plate overnight. The cells were then transfected with plasmids using the Lipofectamine 2000 reagent (11668019, Thermo Scientific, Waltham, MA, USA). Cells were harvested 24–48 h after transfection for further experiments.

### 2.4. Quantitative Reverse Transcription-Polymerase Chain Reaction (qRT-PCR) Analysis

The cochlear tissues of the SD rats were transferred to DNase/RNase-free microcentrifuge tubes. Total RNA was isolated from the homogenates using a Total RNA Kit I (R6834-01, Omega Bio-Tek, Norcross, GA, USA) following the manufacturer’s recommendations. The RNA was quantified using a spectrophotometer (NanoQ, CapitalBio Technology, Beijing, China). Total RNA was reverse transcribed into complementary DNA (cDNA) using a PCR thermocycler (MJ Mini Personal Thermal Cycler; Bio-Rad Laboratories, Hercules, CA, USA) and RevertAid First Strand cDNA Synthesis Kits (K1622, Thermo Fisher Scientific, Waltham, MA, USA). Next, 11 μL of total RNA dissolved in RNase-free water, 1 μL of Oligo (dT)18 primer, 4 μL of 5 X reaction buffer, 1 μL of RiboLock RNase inhibitor, 2 μL of dNTPs mix, then 1 μL of RevertAid reverse transcriptase was mixed and incubated at 42 °C for 60 min and inactivated at 70 °C for 5 min.

qRT-PCR analysis was performed on a CFX96 Real-Time System (Bio-Rad) using SYBR Premix Ex Taq II (RR820A, TaKaRa Biotechnology, Shiga, Japan). The 12.5 μL reaction mixture contained 1 μL of cDNA template, 0.5 μL of each primer, 4.25 μL RNase-free water, and 6.25 μL of SYBR Premix Ex Taq II. The reaction consisted of 95 °C for 30 s followed by 40 cycles of 95 °C for 5 s and 60 °C for 30 s. All qRT-PCR reactions were performed in triplicates, and the resulting values were combined into a mean cycle threshold. The DDCT method, with GAPDH as the endogenous reference, was used to determine the relative levels of gene expression. All primer sequences are presented in the Appendix A.

### 2.5. Western Blot Analysis

Cochlear tissues were thoroughly ground and transferred to lysis buffer containing 1% phosphatase inhibitors, protease inhibitors, and phenylmethylsulfonyl fluoride. The samples were then centrifuged at 12,000× *g* and 4 °C for 30 min. The supernatant was drained into new EP tubes and quantified using a BCA Protein Assay Kit (13222, Cowin bio, Jiangsu, China). Each sample was then loaded into the respective lanes of the gel, separated by 10% sodium dodecyl sulfate-polyacrylamide gel electrophoresis and transferred onto a polyvinylidene fluoride membrane (IPVH00010, Millipore, Burlington, MA, USA). The blots were incubated for 1 h in blocking buffer containing 5% non-fat dry milk in 0.1% Tween 20 in phosphate buffer solution (PBS-T) and then incubated overnight at 4 °C with the corresponding primary antibody diluted in blocking buffer. The blots were washed in PBS-T, incubated for 2 h at 25 °C with a peroxidase-conjugated secondary antibody and developed using an enhanced chemiluminescence reagent (34577, Thermo Fisher Scientific, Waltham, MA, USA). Immunoreactive bands were visualized using a chemiluminescence system (e-BLOT; Touch Imager, Beijing, China).

The antibodies used for western blotting included: polyclonal rabbit anti-BNP antibody (1:500; PA5-96084, Thermo Fisher Scientific, Waltham, MA, USA), polyclonal rabbit anti-NPRA antibody (1:500; PA5-29049, Thermo Fisher Scientific, Waltham, MA, USA), monoclonal mouse anti-rabbit Vitamin D Receptor antibody (1:300; 67192, Proteintech Group, Hubei, China), monoclonal rabbit anti-Vitamin D Receptor antibody (1:500; ab109234, Abcam, Cambridge, UK), polyclonal rabbit anti-GAPDH antibody (1:1000; 10494-1-AP, Proteintech Group, Hubei, China), HRP goat anti-mouse IgG(1:2000; 15014, Proteintech Group, Hubei, China), and HRP goat anti-rabbit IgG (1:2000; 15015, Proteintech Group, Hubei, China).

### 2.6. Preparation of Cochlea Section and Immunofluorescence

The cochleae of rats were perfused and fixed with 4% paraformaldehyde (PFA) through round and oval windows and then incubated overnight at 4 °C. The cochleae were decalcified in 5% EDTA solution for 2 days, frozen overnight in 30% sucrose solution at 4 °C, embedded in a tissue-Tek OCT compound (4583, Sakura Finetek, Japan) at −20 °C, sectioned into 10 μm thick sections using a cryostat microtome (CM1950, Leica, Germany), and attached on poly-L-lysine-coated slides.

The antibodies used in immunofluorescence included polyclonal rabbit anti-BNP antibody (1: 400; PA5-96084, Thermo Fisher Scientific, Waltham, MA, USA), polyclonal rabbit anti-NPRA antibody (1: 500; PA5-29049, Thermo Fisher Scientific, Waltham, MA, USA), monoclonal mouse anti-tubulin β-III primary antibody (1:400; ab78078, Abcam, Cambridge, UK), polyclonal chicken anti-tubulin β-III antibody (1:500; GTX85469, GeneTex, Irvine, TX, USA), Alexa Fluor 488 conjugated donkey anti-mouse IgG (1: 400; A-21202, Thermo Fisher Scientific, Waltham, MA, USA), Alexa Fluor 594-conjugated donkey anti-rabbit IgG (1: 400; A-21207, Thermo Fisher Scientific, Waltham, MA, USA), and Alexa Fluor 647-conjugated goat anti-chicken IgY (1:400; ab150176, Abcam, Cambridge, UK). The samples were fixed with 4% PFA for 20 min, then washed with PBS three times for 5 min each time, soaked with 1% Triton X-100 for 10 min, washed with PBS three times, and then incubated with 5% bovine serum albumin (BSA; v900933-100G, Sigma-Aldrich, St. Louis, MO, USA) for 45 min at 37 °C. The antibodies were diluted and added to the samples according to the manufacturer’s instructions. After incubation at 4 °C for 24 h with the corresponding primary antibody, the samples were washed with PBS three times, incubated with the Alexa Fluor 488 conjugated donkey anti-mouse IgG and Alexa Fluor 594-conjugated donkey anti-rabbit IgG or Alexa Fluor 647 conjugated goat anti-chicken IgY for 2 h at 25 °C, then rinsed with PBS three times, followed by DAPI for 10 min.

### 2.7. Spiral Ganglion Explants Cultures

A P0 rat cochlea was immersed in cold HBSS, the cochlear capsule was opened using fine tissue forceps, and the membranous labyrinth was removed from the volute under an anatomical microscope. The middle turn of the SG was carefully separated from the spiral plate, cut into equal 300 to 500 μm sections, and transferred to 15 mm culture dishes, which were previously coated with the Cell-Tak Cell and Tissue Adhesive (Corning, 354240), and precoated 15 mm glass bottom culture dishes (NEST, 801002), and loaded with 100 μL attachment medium consisting of DMEM, 10% FBS, 25 mM HEPES buffer, and 1% penicillin-streptomycin (all Thermo Fisher Scientific, Waltham, MA, USA). Then, 100 μL of 20% Matrigel (356234, Corning, New York, NY, USA) and DMEM mixture were added to each dish for a three-dimensional culture, and the tissues were left adhering overnight at 37 °C, 5% CO_2_, and 95% humidity (Appendix A). After adherence, the SG explants cultured in neuro maintenance medium with or without 20 ng/mL recombinant BDNF served as controls. The experimental culture was supplemented with 1 μM BNP (RP11121, Caymanchem, Ann Arbor, MI, USA), 1 μM membrane-permeable cGMP analog 8-(4-chlorophenylthio), guano-sine-3′,5′-cyclic monophosphate (8-pCPT-cGMP; Sigma-Aldrich, C5438), 1 μM BNP plus 1 μM PKG inhibitor KT5823 (420321, Sigma-Aldrich, St. Louis, MO, USA), or 1 µM BNP plus 1 µM NPR-A antagonist A71915 (4030385, Bachem, Switzerland) in nerve maintenance medium. In each condition, three cochlear nerve explants were cultured in a wet incubator containing 5% CO_2_ at 37 °C for 7 days, and the medium was changed every other day. The SG explants were fixed with 4% PFA at 25 °C for 20 min on the last day, followed by a neurite growth study.

### 2.8. Spiral Ganglion Neurons Cultures

As described previously, a dissociated culture of SGNs was isolated from P3 newborn rats [10,14,15]. In icy HBSS, every SG was separated from the cochlea by sequentially removing the bony cochlear capsule, spiral ligament, and organ of Corti, leaving the SGNs in the modiolus. These nerve tissues then moved to Ca2+/Mg2+-free HBSS, which contained 0.25% trypsin and 0.1% collagenase type IV (all Thermo Fisher Scientific, Waltham, MA, USA), at 37 °C for 20 min to enzymatically dissociate the cells. Furthermore, 10% FBS was added to quench the enzymatic reaction. After three washes with the medium, the ganglion was separated using a grinding machine with a 1 mL mechanical pipette. The dissociated cells were plated in a prepared culture medium consisting of DMEM/Ham’s F12 medium supplemented with 1× B27, 1× N2, and 1% penicillin-streptomycin (all Thermo Fisher Scientific, Waltham, MA, USA) and dripped into a culture dish previously coated with poly-L-lysine (0.1 mg/mL in 10 mM borate buffer, pH 8.4; Thermo Fisher Scientific, Waltham, MA, USA) to adhere for 4 h at 37 °C, 5% CO_2_, and 95% humidity (Appendix A). Next, 1 mL of neural maintenance medium supplemented with 1 μM BNP was added to the experimental dishes. Culture dishes with or without 20 ng/mL recombinant brain-derived neurotrophic factor BDNF (AF-450-02, Pepro Tech, Waltham, MA, USA) were used as controls. The culture medium was changed every other day, followed by fixation with 4% PFA at 25 °C for 20 min on the last day.

### 2.9. Cochlear Explants Cultures

P3-P4 rats were decapitated, and the cochlea was immersed in cold HBSS. The cochlear capsule was then opened with fine tissue forceps, and the membranous labyrinth was removed from the volute under an anatomical microscope. The middle turns of the cochlear explants containing SGNs were then transferred to 10 mm culture dishes that were previously coated with Cell-TaK and incubated in DMEM supplemented with 10% fetal bovine serum and 1% penicillin-streptomycin at 37 °C under a 5% CO_2_ atmosphere.

The next day, the medium was first incubated with 3 × 10^8^ TU/mL LV-VDR for 24 h and replaced with normal medium for another 48 h at 37 °C and 5% CO_2_. Cochlear explants were harvested, and total protein was obtained and prepared for western blotting.

### 2.10. Immunofluorescence of SG Explants and SGN Cell Neurite Growth

After the culture period, the SG explants or SGN cells were fixed in 4% PFA for 20 min at RT. The explants and cells were enclosed in PBS with 5% BSA and 0.1% Triton X-100, and then incubated with 1:500 anti-tubulin β-III primary antibody, followed by incubation with 1:500 Alexa Fluor 488 conjugated donkey anti-mouse IgG. The nuclei were visualized using DAPI (diluted 1:1000).

### 2.11. Neurite Tracing and Counting

In vitro, images of the immunostained cultures were analyzed using ImageJ software (NIH, version 1.46r) according to a previous study [16]. Neurite tracing was performed by using the “Analyze-Set Scale” function, with the pixel unit of neurite length measurement set in micrometers. Images were rendered with a segmentation function, and a neurite tracer function was applied by choosing the starting point at the SGN cell bodies, resulting in a compiled skeleton rendering of all measured neurites. Only neurites that were fully included in the images were analyzed. The neurite growth of the SG explants was assessed by measuring the number and length of SG28Ns. The total number and neurite length of dissociated SGNs were also analyzed.

### 2.12. CHIP Assays

ChIP assays were performed using the ChIP assay kit (ab500, Abcam, Cambridge, UK). According to the manufacturer’s instructions, 293T cells were cross-linked with formaldehyde and sonicated to obtain 200–300 bp DNA fragments. These fragments were then enriched and precipitated via antigen antibody-specific binding reactions. ChIP was performed using a chip-grade antibody directed against VDR (ab109234, Abcam, Cambridge, UK). The cross-linking between protein and DNA was then removed, the protein was isolated, and the DNA was purified. PCR was performed using primers for VDR-binding sites. Total nuclear-extracted DNA was used as a PCR input control, and PCR products were analyzed by gel electrophoresis on a 2% agarose gel. All primer and promoter sequences are presented in the Appendix A.

### 2.13. Luciferase Assay

Briefly, wild-type BNP 3′-UTR segments were PCR amplified and inserted into the pGL3-Basic vector to generate wild-type BNP plasmids. Mutant 3′-UTR segments of BNP containing mutated sequences at VDR complementary sites were generated by site-directed mutation of the wild-type plasmid. Luciferase activity was calculated as the ratio of firefly to Renilla luminescence.

### 2.14. Zebrafish Husbandry and Imaging

Tg (elavl3: EGFP) zebrafish were obtained from the Zebrafish International Resource Center. By using the promoter of the neurodevelopmental marker gene elavl3 to drive EGFP expression, Tg (elavl3: EGFP) zebrafish can specifically express green fluorescence in neurons from birth. We crossed zebrafish breeders and isolated transparent larvae. All the zebrafish larvae were randomly divided. Briefly, zebrafish larvae were divided into six groups and distributed in 6-well plates treated with 10 nM vitamin D (IC0306, Solarbio, Beijing, China), 1 μM BNP, 1 μM BNP plus 1 μM KT5823, 1 μM BNP plus 1 μM A71915, 1 μM 8-pCPT-cGMP, and no supplement with the fluid changed every day. Images of larval zebrafish embedded in 3% methylcellulose were taken using a microscope (AXIO Zoom.V16, ZEISS, Germany).

### 2.15. Statistical Analysis

Statistical analysis was performed using a one-way analysis of variance followed by Bonferroni’s post hoc test or t-test. Data presented in the text and figures are the means and standard errors of the mean (mean ± SD). Statistical analyses were performed using GraphPad Prism 8 and SPSS 23.0. *p*-values less than 0.05 (*p* < 0.05) were considered to indicate significance.

## 3. Results

### 3.1. BNP Is Expressed in SGNs, Promoting Neuronal Survival and Neurite Growth

In cochlear sections, the BNP immune response appeared in β-III tubulin-positive SGNs, and the distribution of BNP remained unchanged throughout development (Figure 1A). We first discovered the expression of BNP in the SG of the rat cochlea. Real-time quantitative RT-PCR products amplified from mRNA extracted from rat cochlea tissue showed expression levels of BNP mRNA (Figure 1D). Western blotting revealed the presence of protein products (Figure 1B,C).

To verify the effect of BNP on SGNs, SG explants from P0 rats were incubated in different culture media with 1 μM BNP, 20 ng/mL BDNF, and no supplement (control), and they cultured in vitro for 7 days. We then counted the neurites of the explants and measured the length of the neurites in each group. In the control group, the average number of neurites per explants was 19.0 ± 2.6, and the average neurite length per explant was 1045 ± 136.1 μm. The number of neurites and neurite growth increased significantly (62.8 ± 6.4 and 1739 ± 98.0 μm, respectively) in cell cultures treated with 20 ng/mL BDNF. Therefore, we used the culture medium with 20 ng/mL BDNF as a positive control. The number and length of neurites were 29.8 ± 3.4 and 1504 ± 318.6 μm, respectively, for 1 μM BNP-treated SG explants (Figure 2A). We observed similar results in isolated SGNs. In the control group, the average number of neurons per culture dish and the average neurite length per dish were 28.7 ± 1.5 and 62.0 ± 12.1 μm, respectively. The average number of neurons per culture dish and the average neurite length per dish in the BDNF group were 50.0 ± 6.5 and 212.6 ± 21.3 μm, respectively. The number of neurites increased (49.0 ± 2.6), and neurite growth lengthened (157.3 ± 28.9) in cell cultures treated with 1 μM BNP (Figure 2B).

To further verify the function of BNP in vivo, we cultured zebrafish larvae with fluorescently labeled neurons in 1 μM BNP and no supplement medium to observe neuronal development in each group. Neuron counting revealed that zebrafish larvae cultured in 1 μM BNP (23.7 ± 3.5) had significantly more developed neurons than the control zebrafish larvae (10.3 ± 1.5) (Figure 2C). These in vitro and in vivo results were significantly different from those of the control group and confirmed that BNP could promote the growth of SG plants, including promoting neuronal survival, increasing neurite number, and elongating neurite growth length (Figure 2D–G).

### 3.2. BNP Promotes Neurite Growth in SG Explants through the NPRA-cGMP-PKG Signaling Pathway

To further explore the mechanism of the effects of BNP on neurites and determine whether BNP promotes neurite growth by binding to NPRA and increasing the concentration of cGMP, we revealed the localization of NPRA in the cochlea and verified the pathway in SG explants. In cochlear sections, the NPRA immune response appeared in β-III tubulin-positive SGNs, consistent with the distribution of BNP, and remained unchanged during the entire development process (Figure 3A). Protein products in western blotting also confirmed our observations (Figure 3B,C). In a follow-up experiment, SG explants were incubated in a culture medium with 1 µM BNP, 1 μM 8-pCPT-cGMP, 1 µM BNP plus 1 µM NPR-A antagonist A71915, 1 µM BNP, and 1 µM PKG inhibitor KT5823, without any supplement. Quantitative analysis of β-III tubulin-positive nerve fibers showed that the number of neurites per explant was 11.5 ± 3.4, and the average neurite length was 1022 ± 134.3 μm for control samples (Figure 3D). Following 1 µM BNP treatment, the average number and length of neurites in the positive control group were 25.8 ± 2.4 and 1300 ± 88.2 μm, respectively. Compared with the control group, a significant increase was observed in neurite number and growth (Figure 3E,F). In the group treated with 1 μM 8-pCPT-cGMP, the number and length of neurites were 26 ± 3.5 and 1739 ± 98.0 μm, respectively, which were also significantly different from those in the controls. After 1 µM of BNP plus 1 µM of NPR-A antagonist A71915 treatment, the number and length of neurites were 19.8 ± 2.2 and 1021 ± 105.1 μm, respectively. In the culture dishes with 1 µM BNP and 1 µM PKG inhibitor KT5823, the resulting number and length of neurites were 18.8 ± 3.0 and 982.2 ± 219.1 μm, respectively (Figure 3D). There was no significant difference compared with the control (Figure 3E,F).

In zebrafish, we found that the average number of neurons in the group with 1 µM BNP (25.0 ± 1.0) was significantly higher than that in the control group (13.7 ± 1.6). The average number of neurons in the group with 8-pCPT-cGMP was 24.7 ± 2.1, which had the same effect as the BNP group. In the group treated with 1 µM BNP plus 1 µM NPRA antagonist A71915 and 1 µM BNP and 1 µM PKG inhibitor KT5823, the average number of neurons was 18.7 ± 1.6 and 17.7 ± 2.5, respectively. There was no significant difference compared with the control (Figure 4A,B).

### 3.3. BNP Is Regulated by Transcriptional Factor VDR

To identify molecules that may interact with *BNP*, we used the *PROMO* website to predict candidate transcription factors and transcription factor-binding motifs. By screening for potential factors and reviewing the literature, BNP was proposed as possibly being directly regulated by VDR. In cochlear sections, we observed VDR expression in SGNs, which overlapped with the distribution of BNP (Figure 5A). To further investigate the relationship between the two, 293T cells were transfected with a VDR-overexpression plasmid. Cells were harvested 24–48 h after transfection for RT-qPCR and western blotting analyses. RT-qPCR showed that VDR overexpression increased BNP levels (Figure 5B). The protein products indicated that the expression of BNP increased with the overexpression of VDR (Figure 5D–F). In rats, cochlear explants were infected with GFP-tagged VDR-overexpressing lentivirus. Green fluorescence overlapped with β-III tubulin-positive SGNs, indicating overexpression of VDR in SGNs (Figure 5C). After 48 h, cochlear explants were harvested for western blotting. The results demonstrated that the level of BNP tended to increase with an increased level of VDR in the rat cochlea (Figure 5D–F). To further illustrate this relationship, we predicted the possible binding sites of VDR on the BNP gene (NPPB) on the *JASPAR* website (http://jaspar.binf.ku.dk/, accessed on 14 April 2021) and selected two sites with the highest predicted scores for chromatin immunoprecipitation (ChIP) analysis (Figure 6A,B). Sequence analysis of the BNP promoter identified putative binding sites for VDR at −992 and −1122 (Figure 6C), suggesting that VDR regulates the transcription of BNP. In ChIP analysis, we obtained 200–700 bp DNA fragments by ultrasonic treatment (Figure 6D), and then enriched and precipitated these fragments through an antigen-antibody-specific binding reaction. The sequence information of the binding sites was obtained by PCR. The results showed that VDR directly binds to the two predicted binding sites of the BNP promoter (Figure 6E,F). Subsequently, we examined the binding of VDR to BNP using a luciferase assay. When 293 control cells were transfected with a luciferase reporter plasmid containing the BNP promoter sequence, an abundant luciferase signal was detected. After the transfection of 293 cells overexpressing VDR, luciferase activity was significantly enhanced. The luciferase signal remained unchanged after mutation of the promoter-binding site sequence (Figure 6G). Similar results were obtained in the rat luciferase reporter assay (Figure 6H,I).

### 3.4. Vitamin D Promotes Neuronal Development in Zebrafish Larvae

Jillanne et al. [17] reported that vitamin D_3_ (10 nM) significantly increased the rate of neurite growth in hippocampal explants. Considering that vitamin D_3_ increases the expression of *VDR* and mediates *VDR* signaling [18], we cultured SG explants and zebrafish larvae in 10 nM vitamin D_3_ to determine whether it would have any effect. As displayed in Figure 7A, the average length and number of neurites in control plants cultured with no supplement were 1167 ± 56.9 µM and 14.3 ± 4.0, respectively. The average length and number of neurons in the 10 nM vitamin D_3_ group were 1676 ± 175.4 µM and 23.3 ± 3.1, respectively, which were significantly different from those in the control group (Figure 7C,D). The number of neurons in zebrafish larvae was 71.3 ± 5.1, which was significantly different from the number in the control (17.7 ± 2.5) (Figure 7E). The SG explants cultured in 10 nM vitamin D plus 1 µM NPR-A antagonist A71915, or 10 nM vitamin D plus 1 μM PKG inhibitor KT5823 proved the same. The length and number of neurites were 1137 ± 79.2 μm and 18.7 ±3.1, respectively, for 10 nM vitamin D plus 1 µM A71915-treated SG explants. The average length and number of neurites in explants cultured with 10 nM vitamin D plus 1 µM KT5823 were 1198 ± 51.7 µM and 19.0 ± 2.0, respectively. In these two groups, the results showed no significant difference in the number and length of neurites compared with the control group (1149 ± 71.2 μm and 21.0 ± 2.6). The same results were observed in zebrafish. The average number of neurons in zebrafish larvae cultured in 10 nM vitamin D plus 1 μM A71915 and 10 nM vitamin D plus 1 µM KT5823 were 20.3 ± 2.1 and 18.3 ± 2.5. There was no significant difference compared with the control (20.7 ± 3.1). The supplements of A71915 and KT5823 inhibited the GMP-PKG pathway in which BNP plays a role, and the pro-neurogenic effects of vitamin D were lost. This part of the figures has been included as a Appendix A. After infection with VDR-overexpressing lentivirus, we also found that the morphology of neurons was more complete than that of untreated explants (Figure 7B).

### 3.5. VDR Regulates BNP Neurite Growth and Survival of SGNs through cGMP-PKG Signaling Pathway

In the rat cochlea, we first determined the expression of BNP and VDR in the SGNs. With the culture of SG explants and single neurons, we demonstrated that BNP could promote neurite growth and survival of SGNs in rat cochlea. Further experiments showed that BNP plays a neuroprotective role by binding to NPRA, increasing the level of the second messenger (cGMP), and activating the downstream substrate PKG. Subsequently, by predicting and elucidating the transcriptional regulation of BNP by VDR in a variety of assays, we completely dissected the role of BNP in the inner ear (Figure 8).

## 4. Discussion

As a cardiac hormone, BNP plays a key role in the regulation of blood pressure and fluid volume [19]. In the early days, BNP was discovered as a biomarker for the identification of congestive heart failure patients, and later studies revealed that high levels of BNP in cardiac tissues indicate cardiac hypertrophy or atrial fibrillation [20,21,22,23]. BNP also plays a vital physiological role outside the heart as mentioned previously. Hormones and secretory proteins have been implicated in a variety of inner ear and hearing disorders, but to the best of our knowledge, the role of BNP in hearing has not been reported. Regarding the rat’s inner ear, our research confirms that BNP plays a significant role in the auditory system. We first explored the natural expression and location of BNP and its receptors and demonstrated that BNP was expressed in the rat SG by real-time quantitative RT-PCR and western blotting. Owing to its localization in SGNs, we explored the effect of BNP on the number of neurons and neurites and found that BNP could promote the survival of neurons and prolong the length of neurites in vitro and in vivo. The experimental data provide direct evidence for the expression and function of BNP in primary auditory neurons of the rat cochlea, demonstrating that BNP may play an important role in regulating the neural function of SGNs through the NPRA-cGMP-PKG signaling pathway.

The expression of genes is often regulated at the transcriptional level by specific transcription factors, which is the most common mode of regulation [24]. Transcriptional regulation is the main step in regulating the expression of most functional protein-coding genes. As a functional protein, BNP is regulated by several transcription factors. JASPER and PROMO websites predicted that VDR is a transcription factor of BNP, and VDR-binding sites in this factor were suggested. It was somewhat surprising that BNP transcription could be regulated by VDR, which was activated by 1,25(OH)_2_D_3_. The ChIP method was used to identify the VDR binding sites. We confirmed that VDR could bind to special sequences of the BNP gene (NPPB) promoter and regulate gene transcription. In cochlear explants and 293T cells, BNP overexpression was observed in conjunction with VDR overexpression when infected with the lentiviral vector overexpressing the VDR gene, and an increase in BNP at both RNA and protein levels was observed. Luciferase reporter assays provided further evidence of their interaction. To the best of our knowledge, this is the first report of this interesting transcriptional regulatory relationship. Through positive regulation of VDR, BNP can further promote neurite growth and neuronal survival. Although there is a lack of understanding of the complicated effects of different factors on the transcriptional regulation of a gene [25], our research remains at the forefront of therapeutic research on SNHL and offers novel approaches for current therapy.

As a result of its binding to vitamin D-responsive elements, VDR modulates the transcription of vitamin D-regulated genes, affecting both physiological and pathological functions [26]. It functions as a crucial genomic agent for vitamin D, regulating the transcription of numerous genes involved in cell proliferation, differentiation, apoptosis, angiogenesis, inflammation, and metastasis, among other processes [27]. Previous research has demonstrated that VDR-deficient animals age prematurely, suggesting that VDR signaling may have anti-ageing benefits [28]. In neurons and skeletal muscles, it has been reported that VDR KO mice had a smaller peripheral nerve axonal diameter and disordered acetylcholine receptor morphology in the extensor digitorum longus muscle. VDR signaling regulates neuromuscular maintenance and enhances the development of nerve axons [29]. A previous animal study showed that VDR-knockout mice display age-related hearing loss in the inner ear [13]. In contrast, 1,25(OH)_2_D_3_ induces nerve growth factors in embryonic rat hippocampal neurons and promotes neurite outgrowth [17]. However, the molecular mechanism of VDR in the inner ear has not been elucidated to the best of our knowledge. Based on the above studies, we further explored the biological function of VDR in the growth and development of SGNs. We found that VDR plays a protective role in the inner ear by regulating factors that promote neurite growth and the survival of cochlear SGNs. It is possible that more than one pathway leads to this protection; however, further research is required. Notably, our results suggested that vitamin D may act as a potential therapeutic agent for SNHL by exerting downstream effects through VDR activation. These findings may have implications for clinical practice, but further research is required to test this hypothesis. In further studies, we will aim to re-recognize the function of the VDR and obtain essential implications for future practice.

## 5. Conclusions

In summary, we described the expression of BNP in the rat’s inner ear and explored its effect on the number and length of neurons and neurites at the cellular and tissue levels. BDNF is a neurotrophic protein that promotes neurite growth [30]. By comparing the growth of isolated neurons and SG explants treated with BNP and BDNF, we found that BNP promoted nerve growth and elucidated the related mechanism. Experiments in zebrafish larvae demonstrated a similar function for BNP, further verifying our results in vivo. In addition, our results indicated a potential novel transcriptional factor. In the search for molecules that may interact with BNP, we predicted and experimentally verified that VDR could bind to BNP and regulate its expression. Thus, we may confirm the effects of BNP and further improve the survival and neurite length of SGNs. The regeneration of SGNs in the auditory system has been the focus of the treatment of SNHL, as SGNs play an essential role in the hearing process, and their peripheral and central processes connect the sensory cells of the Corti to the central nervous system [31]. Collectively, our findings may be a preliminary study to examine the role of BNP and VDR in the ear, providing a novel idea for neuronal regeneration therapy. Their role in noise-induced and age-related hearing loss warrants further investigation.

## Figures and Tables

**Figure 1 cells-11-03746-f001:**
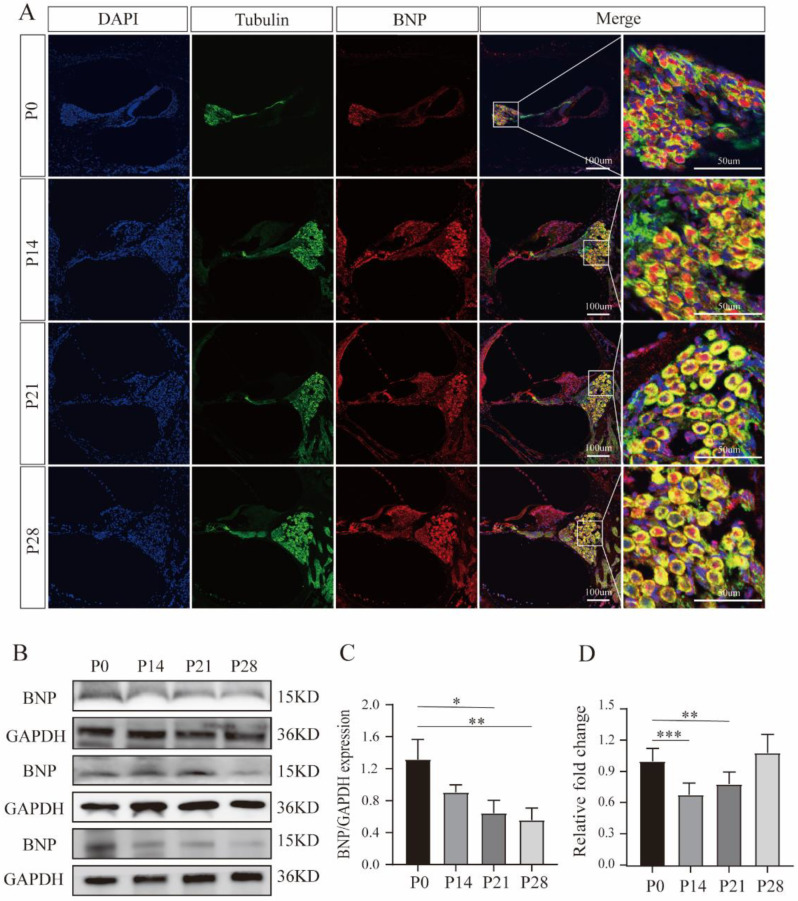
Expression and localization of brain natriuretic peptide (BNP) in the cochlea. (**A**) BNP is expressed in rat spiral ganglion neurons and remains unchanged during the whole development. (**B**) BNP protein expression at different ages detected by Western blot, using GAPDH as the endogenous housekeeping control gene. (**C**) Histograms for Western blot results, fold change relative to P0. (**D**) Relative quantification of expression of BNP in the rat spiral ganglion using GAPDH as the endogenous housekeeping control (fold change relative to P0). All of the data are presented as the mean ± SD of three independent experiments. *, *p* < 0.05, **, *p* < 0.01, ***, *p* < 0.001 compared with the P0.

**Figure 2 cells-11-03746-f002:**
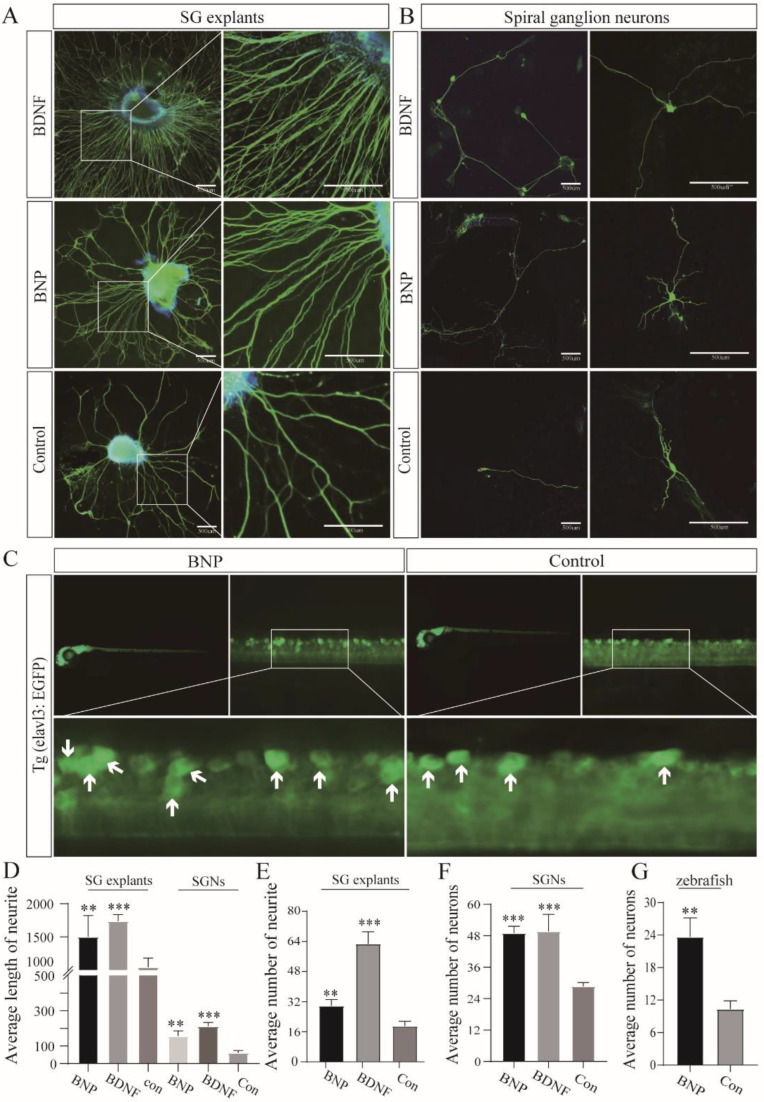
BNP promotes neurite growth and survival of cochlear spiral ganglion neurons. (**A**) Immunofluorescence analysis of neurite length and number of spiral ganglion (SG) explants cultured in 20 ng/mL BDNF, 1 µM BNP and no supplement. Scale bar: 500 μm. (**B**) The number of neurons and the length of neurites were observed by fluorescence immunostaining. Scale bar: 500 μm. (**C**) Neuron number counting of zebrafish with fluorescently labeled neurons cultured in 1 µM BNP group and the control group. Arrows indicate complete and countable neurons. (**D**) Left, comparison of neurite length between SG explants cultured in 20 ng/mL BDNF, 1 µM BNP and no supplement medium. Right, comparison of neurite growth length of single neuron cultured in 20 ng/mL BDNF, 1 µM BNP and no supplement. (**E**) The number of neurites in SG explants was compared under three culture conditions. (**F**) Comparison of the number of single neuron cultures three different conditions. (**G**) Comparison of neuron counts in zebrafish cultured in 1 µM BNP and no supplement medium. All of the data are presented as the mean ± SD of three independent experiments. **, *p* < 0.01, ***, *p* < 0.001 compared with the control.

**Figure 3 cells-11-03746-f003:**
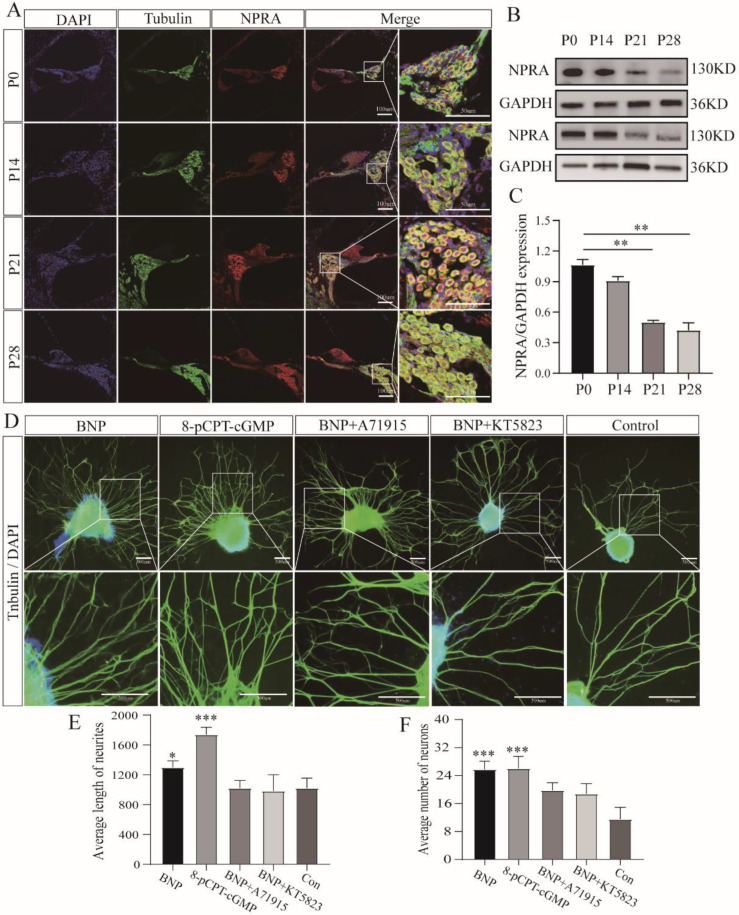
BNP promotes neurite growth in spiral ganglion explants through the NPRA-cGMP-PKG Signaling pathway. (**A**) Expression and localization of NPRA in the cochlea. NPRA is expressed in rat spiral ganglion neurons and remains unchanged throughout development. (**B**) NPRA protein expression at different ages detected by Western blot, using GAPDH as the endogenous housekeeping control gene. (**C**) Histograms for Western blot results, fold change relative to P0. (**D**) SG explants were maintained in culture medium alone or medium supplemented with 20 ng/mL 1 µM BNP, 1 µM 8-pCPT-cGMP, 1 µM BNP plus 1 µM A71915, 1 µM BNP plus 1 µM KT5823 for 5 days. SG plants were double-labeled with TuJ1 antibody (green) and Hoechst (blue). Scale bar: 500 μm. (**E**) The average neurite growth length was quantified and compared under different conditions. (**F**) The average number of neurites was compared among five groups. All of the data are presented as the mean ± SD of three independent experiments. *, *p* < 0.05, **, *p* < 0.01, ***, *p* < 0.001 compared with the control.

**Figure 4 cells-11-03746-f004:**
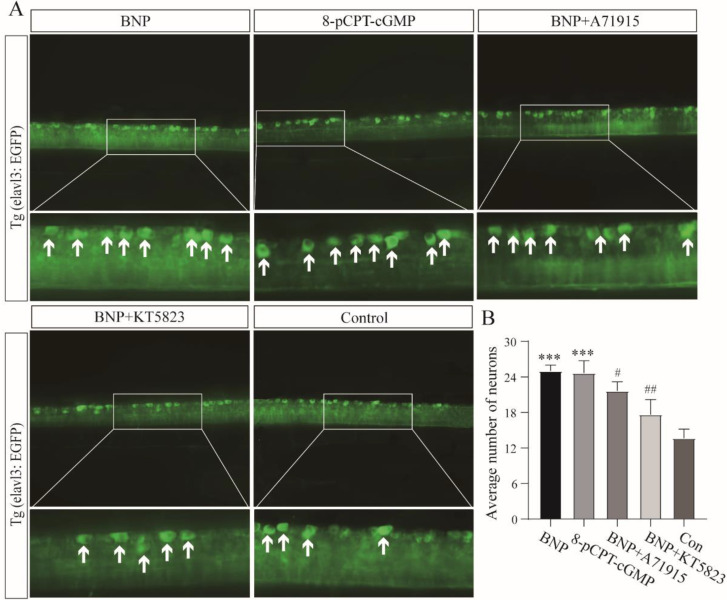
BNP promotes neurite growth in vivo through the NPRA-cGMP-PKG Signaling pathway. (**A**) Zebrafish larvae were maintained in a culture medium alone or a medium supplemented with 1 µM 8-pCPT-cGMP, 1 µM BNP plus 1 µM A71915, 1 µM BNP plus 1 µM KT5823 for 2 days. Arrows indicate complete and countable neurons. (**B**) Comparison of the average number of neurons in different treated zebrafish larva. All of the data are presented as the mean ± SD of three independent experiments. ***, *p* < 0.001 compared with the control. #, *p* < 0.05, ##, *p* < 0.01, compared with BNP group.

**Figure 5 cells-11-03746-f005:**
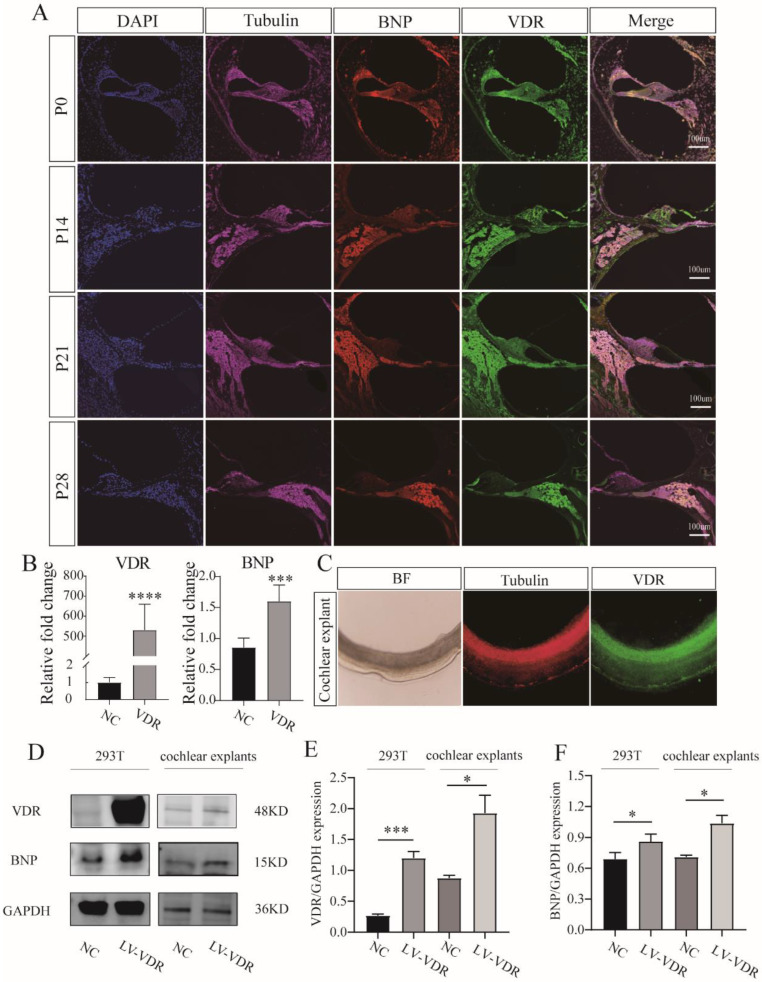
Vitamin D receptor (VDR) expression in rat spiral ganglion neurons and association between VDR overexpression and BNP expression. (**A**) VDR is expressed in rat spiral ganglion neurons and remains unchanged during the whole development, overlapping with BNP. (**B**) PCR analysis of BNP expression after VDR overexpression. Cochlear explants transfected with negative control (NC) lentiviral served as controls. Overexpression of VDR increased BNP RNA levels. (**C**) β-III Tubulin positive spiral ganglion neurons of cochlear explants were infected with GFP-tagged VDR overexpression lentivirus. (**D**) Western blot analysis of BNP at 48 h after transfection in rat cochlear explants and 293 T cells, using GAPDH as the endogenous housekeeping control gene. Cochlear explants transfected with negative control lentiviral served as controls. (**E**,**F**) Histograms for Western blot results show the level of BNP tended to increase with an increased level of VDR in rat cochlear and 293 T cells. All of the data are presented as the mean ± SD of three independent experiments. *, *p* < 0.05, ***, *p* < 0.001, ****, *p* < 0.0001compared with the NC.

**Figure 6 cells-11-03746-f006:**
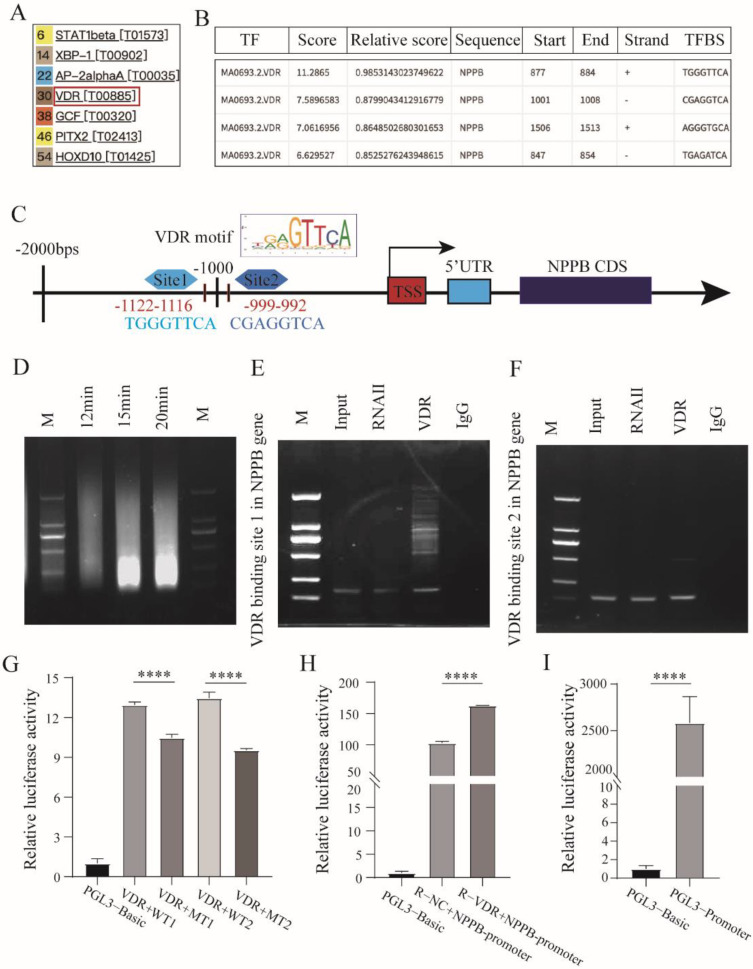
VDR binds to BNP promoters to regulate BNP gene expression. (**A**) Possible BNP binding transcription factors predicted by *PROMO*. (**B**) Predicted VDR binding sites in human BNP(NPPB) gene promoter regions. Human NPPB gene upstream regions were analyzed for VDR binding sites by *JASPAR* online analysis (http://jaspar.genereg.net, accessed on 14 April 2021). (**C**) Schematic diagram of predicted VDR binding sequences within a region 999 bp and 1122 upstream of the BNP gene locus TSS. (**D**) DNA fragments were obtained after different times of ultrasound before ChIP assays. (**E**,**F**) ChIP assays were performed to determine the binding of VDR to the BNP promoter region in 293 T cells. The two regions amplified by the specific PCR primers in ChIP assays are indicated. (**G**–**I**) Luciferase activity assays were conducted to assess the interaction between NPPB gene promoter regions and transcription factor VDR in human 293 T cells and rats. Luciferase reporter plasmid containing NPPB promoter sequence (or blank sequence) were transinfected into the control cells or VDR expressed cells, luciferase activity was quantitatively analyzed. The data are presented as the mean ± SD, ****, *p* < 0.0001.

**Figure 7 cells-11-03746-f007:**
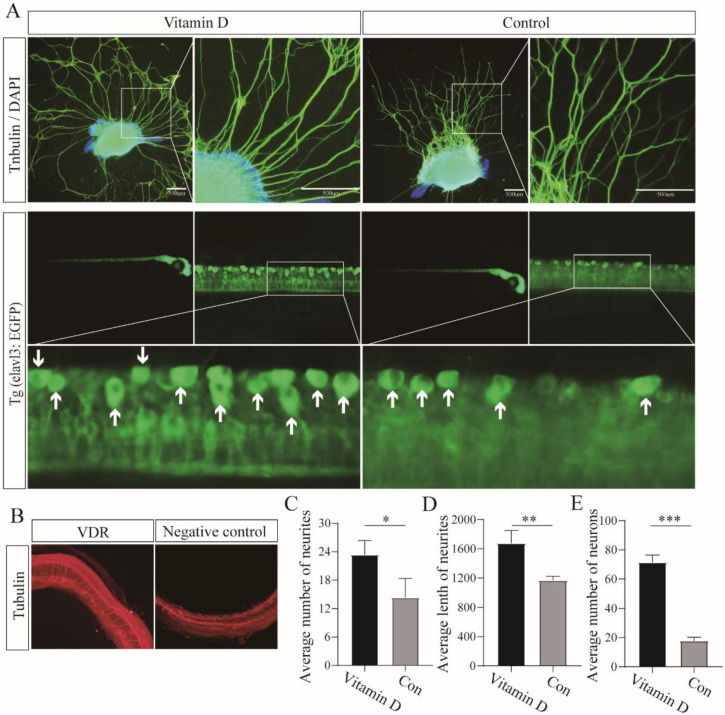
Vitamin D (VD) promotes neuronal survival and neurite growth by binding to VDR and stimulating VDR expression up-regulation. (**A**) Up, SG explants were cultured in 10 nM VD and no added medium for 7 days followed by fluorescence immunostaining analysis. Down, zebrafish larva was maintained in culture medium alone or medium supplemented with 10 nM VD. Arrows indicate complete and countable neurons. (**B**) Fluorescence immunostaining of cochlear explants after transfection with VDR overexpressing lentivirus and negative control lentivirus. (**C**,**D**) Comparison of average neurite growth length and number between VD-supplemented cochlear explants and control cochlear explants. (**E**) Comparison of the average number of neurons in different treated zebrafish larva. All of the data are presented as the mean ± SD of three independent experiments. *, *p* < 0.05, **, *p* < 0.01, ***, *p* < 0.001 compared with the control.

**Figure 8 cells-11-03746-f008:**
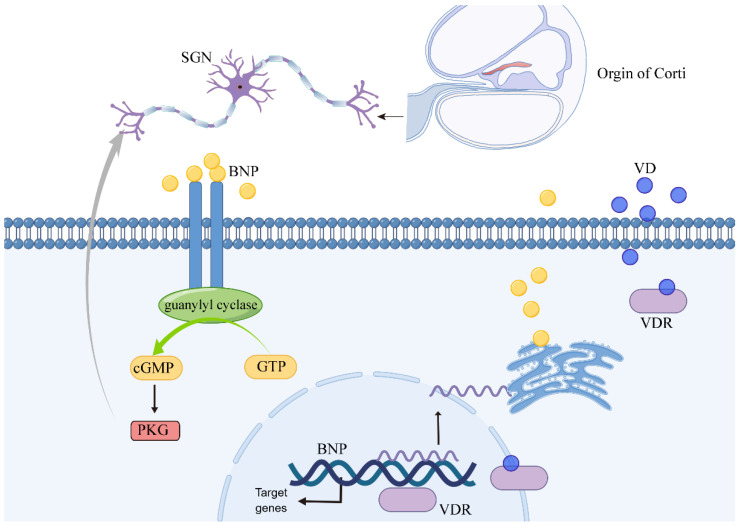
Schematic overview of the role of VDR-BNP in promoting neurite growth and survival of SGNs. Crossing the cell membrane and binding to VDR, VD promotes VDR translocation to the nucleus. In the nucleus, VDR positively regulates BNP expression by binding to BNP gene promoter. Then, BNP plays a neuroprotective role by binding to NPRA, activating guanylyl cyclase and increasing the level of the second messenger (cGMP), then activating cGMP-dependent protein kinase G.

## Data Availability

The data that support the findings of this study are available from the corresponding author upon reasonable request.

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
