# Peer review of "VDR Regulates BNP Promoting Neurite Growth and Survival of Cochlear Spiral Ganglion Neurons through cGMP-PKG Signaling Pathway"

_cells, 2022, doi:10.3390/cells11233746_

Round 1
Reviewer 1 Report
The manuscript “VDR Regulates BNP Promoting Neurite Growth and Survival of Cochlear Spiral Ganglion Neurons Through cGMP-PKG Signaling Pathway” is very interesting and well-written. Data was well-presented. There are some minor problems.
1. Write the aim of this study more clearly in the Introduction section
2. Line 275-Legend Fig 1D: “control protein”
Author Response
Point1: Write the aim of this study more clearly in the Introduction section
Response 1: Thank you very much for taking the time to read our manuscript and for recognising us. We have clarified this study's aim more clearly in the Introduction section
Point2: Line 275-Legend Fig 1D: “control protein”
Response 2: We have carefully deleted the superfluous words and examined the full manuscript. Thanks for your careful reading.

Reviewer 2 Report
In their study entitled “VDR Regulates BNP Promoting Neurite Growth and Survival of Cochlear Spiral Ganglion Neurons Through cGMP-PKG Signaling Pathway” Zhang et al. demonstrate that B-type natriuretic peptide (BNP) is expressed in the rat inner ear spiral ganglion neurons (SGNs) and promotes neuron survival and prolongs neurite length. They also show that the vitamin D receptor (VDR), a transcriptional factor, could upregulate the expression of BNP and enhance its function. There is another paper from the same group published in Frontiers in Cell and Developmental Biology in 2021 showing the same story using same methods for ANP. Since ANP and BNP have similar structures and both bind to natriuretic peptide receptor-A (NPR-A), which is coupled to cGMP production, the same effects and mechanisms are not at all surprising but rather anticipated. Therefore, this part of the study does not offer any novelty. The more interesting part is the observation that BNP is regulated by the transcriptional factor VDR and the identification of two binding sites of VDR on the BNP promoter.
Major comments:
1. It is inferred by the manuscript that the pro-neurogenic effects of vitamin D3 are mediated by BNP upregulation. However, there is no pharmacological intervention blocking the BNP signaling when using vitamin D. An antagonist to NPR-A or a PKG inhibitor should be used should be used with vitamin D to verify this theory.
2. The title of the manuscript specifies cGMP-PKG Signaling Pathway as the transducer of SGN survival and neurite outgrowth; however, in the abstract there is no mention of the pathway and how it links VDR and BNP to such outcomes. Therefore, the abstract should be rewritten to contain this essential part of the study (according to the title). In general, the abstract should be improved to reflect all findings of the study.
Minor comments:
Page 4, line 183 SGN should be spelled out in the subtitle “2.8. SGN culture”.
Author Response
Thank you for this valuable feedback. We are very grateful for your recognition of our previous work and valuable comments on this manuscript. Our responses to your comments are as follows.
Point1: It is inferred by the manuscript that the pro-neurogenic effects of vitamin D3 are mediated by BNP upregulation. However, there is no pharmacological intervention blocking the BNP signaling when using vitamin D. An antagonist to NPR-A or a PKG inhibitor should be used should be used with vitamin D to verify this theory.
Response 1: For blocking the BNP signaling when using vitamin D, SG explants were cultured in 10 nM vitamin D plus 1 µM NPR-A antagonist A71915, or 10 nM vitamin D plus 1 μM PKG inhibitor KT5823. The SG explants were cultured by the same method as in our manuscript, followed by fluorescent immunostaining and a neurite growth study. The length and number of neurites were 1137 ± 79.2 μm and 18.7 ±3.1, respectively, for 10 nM vitamin D plus 1 µM A71915-treated SG explants. The average length and number of neurites in explants cultured with 10 nM vitamin D plus 1 µM KT5823 were 1198 ± 51.7 µM and 19.0 ± 2.0, respectively. In these two groups, the results showed no significant difference in the number and length of neurites compared with the control group (1149 ± 71.2 μm and 21.0 ± 2.6). The same results were observed in zebrafish. The average number of neurons in zebrafish larvae cultured in 10 nM vitamin D plus 1 μM A71915 and 10 nM vitamin D plus 1 µM KT5823 were 20.3 ± 2.1 and 18.3 ± 2.5. There was no significant difference compared to the control (20.7 ± 3.1). The supplements of A71915 and KT5823 inhibited the GMP-PKG pathway in which BNP plays a role, and the pro-neurogenic effects of vitamin D were lost. This part of the figures has been included as a supplemental material file.
Point2: The title of the manuscript specifies cGMP-PKG Signaling Pathway as the transducer of SGN survival and neurite outgrowth; however, in the abstract, there is no mention of the pathway and how it links VDR and BNP to such outcomes. Therefore, the abstract should be rewritten to contain this essential part of the study (according to the title). In general, the abstract should be improved to reflect all findings of the study.
Response 2: We have supplemented all of our findings in the abstract section. In the abstract, we supplement the pathways found in the study and clarify the link between VDR and the pathway in which BNP plays a role.
(We have modified Page 4, line 183 SGN, according to your suggestion)

Reviewer 3 Report
This study examines the role of BNP in promoting neurite growth and survival of SGNs.
Abstract - the authors imply that there will be experimentation involving SNHL, however, none is performed. This is a cellular study on the affects of BNP. The abstract should be clearer by formatting the abstract as Introduction, Methods, Results, and Discussion - that would focus on what was actually performed and analyzed. The authors should explain why they used SD rats - are they sensitive to hearing loss, as some mouse breeds are? Results should be compared to age-appropriate controls, since SNHL is age-related.
Introduction - Line 35. Some would argue that SNHL is caused prior to SGN loss, current thinking is that it is begun by loss of synapses to the SGN neurons. Authors should address this. Authors should address why they chose BNP and why they thought it would be a better choice than other neural-growth promoting hormones that have also been studied.
Please explain abbreviations when they are first introduced, including genes and proteins.
The authors should draw a diagram of the sequential experimentations - the interventions and choice of material, i.e., spiral ganglion explants, zebra fish, etc. are unclear from the introduction.
Discussion - line 483. Although BNP has not been reported, other neural growth hormones have been looked at. Why is this better?
Line 506-508 is redundant.
None of these experiments were performed with any hint of SNHL - how do the authors leap to the conclusion that this would aid in repair?
Figure 8. It would be reasonable to place this figure earlier, so that the reader can understand why these experiments were performed, and the significance of the various proteins.
General - The authors should spell out abbreviations for each figure.
Discussion - Since this article is about spiral ganglion cells, the authors should consider focusing on their role, rather than the uses of BNP in the heard, kidney, and lungs. The authors should clearly state why they chose to explore BNP for hearing loss. How much is naturally seen in the spiral ganglion cells?
In addition, the authors should focus on the role of spiral ganglion cells in hearing loss. The cells themselves do not appear to die until a long time after the synapses to inner hair cells are lost, in fact the cells survive for a long time. In addition, are the authors looking at age-related changes, or trauma? Since there is not intervention for trauma, that would not be part of the article.
Author Response
Point1: Abstract - the authors imply that there will be experimentation involving SNHL, however, none is performed. This is a cellular study on the affects of BNP. The abstract should be clearer by formatting the abstract as Introduction, Methods, Results, and Discussion - that would focus on what was actually performed and analyzed. The authors should explain why they used SD rats - are they sensitive to hearing loss, as some mouse breeds are? Results should be compared to age-appropriate controls, since SNHL is age-related.
Response 1: We mentioned SNHL at the beginning of the manuscript, which led to the importance of spiral ganglion neurons and the necessity of protecting spiral ganglion neurons. Our study found that BNP is expressed in the inner ear and promotes neurite growth and survival of cochlear spiral ganglion neurons, which could be regulated by VDR. We hope our findings will be a reference for research on neuronal regeneration therapy for SNHL. As for the abstract format you mentioned, we have revised the abstract more clearly based on the format required by the journal.
As commonly used tool rats, SD rats reproduce quickly, have slightly larger body sizes and have a better ability to resist diseases. SD rats have modestly-sized cochlea and a nervous system similar to humans. It's more suitable for cochlear and SGNs explants culture and neural research. We explored the protective methods of spiral ganglion neurons and did not directly explore the effect of age on SNHL, so no age-related controls were set in our experiments.
Point2: Introduction - Line 35. Some would argue that SNHL is caused prior to SGN loss, current thinking is that it is begun by loss of synapses to the SGN neurons. Authors should address this. Authors should address why they chose BNP and why they thought it would be a better choice than other neural-growth promoting hormones that have also been studied.
Please explain abbreviations when they are first introduced, including genes and proteins.
The authors should draw a diagram of the sequential experimentations - the interventions and choice of material, i.e., spiral ganglion explants, zebra fish, etc. are unclear from the introduction.
Response 2 : Synapse is an important structure in which SGNs play a role. In future studies, we also want to explore the relationship between SNHL and synaptic loss more comprehensively and try to find more protective mechanisms. BNP plays an important physiological function in various organs as a common hormone in the body. There are many clinical studies on BNP, but few basic ones. In previous work, we have identified the neuroprotective effects of ANP. In this manuscript, we chose BNP as the study object to further explore the function of the natriuretic peptide family in the inner ear.
We have spelt out all abbreviations which were first introduced.
Thanks for your reminder. We have put the sequential experimental diagram into the supplementary materials.
Point3: Discussion - line 483. Although BNP has not been reported, other neural growth hormones have been looked at. Why is this better?
Line 506-508 is redundant.
Response 3: As a common hormone in the body, BNP plays an important physiological function in various organs. To our knowledge, the expression and function of BNP in the inner ear have not been reported. Based on this, we investigated the role of BNP in the inner ear and explored potential regulatory mechanisms.
After careful reading, we have removed the redundant parts.
Point4: None of these experiments were performed with any hint of SNHL - how do the authors leap to the conclusion that this would aid in repair? Figure 8. It would be reasonable to place this figure earlier, so that the reader can understand why these experiments were performed, and the significance of the various proteins.
General - The authors should spell out abbreviations for each figure.
Response 4: SNHL is mainly caused by damage and degeneration of cochlear hair cells and spiral ganglion neurons (SGNs), primary neurons that carry auditory information and transmit signals from hair cells to the auditory centre after initial coding processing. Thus, SGN degeneration accelerates the progression of SNHL. Our study found that BNP can promote spiral ganglion neuron survival and neurite growth, so we consider this as a potential treatment for sensorineural hearing loss.
We had uploaded the graphical abstract of the manuscript before, which described the manuscript's content in a general way, and attached figure 8 at the end of the body to summarize the full text.
We have spelt out all abbreviations which were first introduced in each figure.
Point5: Discussion - Since this article is about spiral ganglion cells, the authors should consider focusing on their role, rather than the uses of BNP in the heard, kidney, and lungs. The authors should clearly state why they chose to explore BNP for hearing loss. How much is naturally seen in the spiral ganglion cells?
In addition, the authors should focus on the role of spiral ganglion cells in hearing loss. The cells themselves do not appear to die until a long time after the synapses to inner hair cells are lost, in fact the cells survive for a long time. In addition, are the authors looking at age-related changes, or trauma? Since there is not intervention for trauma, that would not be part of the article.
Response 5: BNP plays an important physiological function in various organs as a common hormone in the body. We listed the different roles it plays in the various organs in our manuscript. In fluorescent immunostaining, we found that BNP was highly expressed in spiral ganglion neurons, which aroused our interest. In a review of the literature, we noticed that the expression and function of BNP in the inner ear had not been reported, and no further reports were found. In addition, there is also not much basic research on the BNP. Based on this, we investigated the role of BNP in the inner ear and explored potential regulatory mechanisms.
There were no experiments involving synapses in our study, but we also noted the vital role of synapses in the overall auditory pathway. In future studies, we look forward to further investigating the relationship between synapses, trauma, and other age-related factors and spiral ganglion neurons. Thank you for your inspiration.

Round 2
Reviewer 2 Report
The authors have sufficiently addressed my concerns.
Author Response
Thank you again for your recognition of us.
Reviewer 3 Report
The authors chose to defend their manuscript rather than change it according to suggestions.
I really think that this could be a better paper by removing most of the information regarding BNP in the rest of the body, making it clear that it does exist in the cochlea, and thus is more of a "natural" hormone. The authors could make it clearer that this does not include any experimentation involving SNHL, that the research animals were not exposed to noise or aging, that this was a preliminary, initial study to examine the role of BNP in the ear, and that future studies should include hearing testing, etc. The abstract and introduction imply that this is about SNHL, but in fact, it is about BNP and its role in the spiral ganglion neurons, as reflected in the title.
Author Response
Thank you very much for your suggestion. After consideration, we decided to follow your advice and delete unnecessary parts to show our research purpose clearly. We emphasized the natural expression of BNP in the cochlea and deleted the part that implied experimentation involving SNHL. In addition, we also stress that this is a preliminary study of BNP and does not involve noise or age-related hearing loss. Thank you again for your sincere advice.